# Treatment of Thoracic SMARCA4-Deficient Undifferentiated Tumors: Where We Are and Where We Will Go

**DOI:** 10.3390/ijms25063237

**Published:** 2024-03-13

**Authors:** Vito Longo, Annamaria Catino, Michele Montrone, Elisabetta Sara Montagna, Francesco Pesola, Ilaria Marech, Pamela Pizzutilo, Annalisa Nardone, Antonella Perrone, Monica Gesualdo, Domenico Galetta

**Affiliations:** 1Medical Thoracic Oncology Unit, IRCCS Istituto Tumori “Giovanni Paolo II”, 70124 Bari, Italy; a.catino@oncologico.bari.it (A.C.); m.montrone@oncologico.bari.it (M.M.); es.montagna@oncologico.bari.it (E.S.M.); f.pesola@oncologico.bari.it (F.P.); i.marech@oncologico.bari.it (I.M.); p.pizzutilo@oncologico.bari.it (P.P.); a.perrone@oncologico.bari.it (A.P.); m.gesualdo@oncologico.bari.it (M.G.); galetta@oncologico.bari.it (D.G.); 2Unità Operativa Complessa di Radioterapia, IRCCS Istituto Tumori “Giovanni Paolo II”, 70124 Bari, Italy; a.nardone@oncologico.bari.it

**Keywords:** SMARCA4-UT, immunotherapy, clinicopathological features

## Abstract

Recently, the fifth edition of the WHO classification recognized the thoracic *SMARCA4*-deficient undifferentiated tumor (SMARCA4-UT) as a separate entity from conventional non-small cell lung cancer with *SMARCA4* deficiency because of the different clinicopathological characteristics of these two diseases. SMARCA4-UT mainly occurs in young to middle-aged adults and involves a large mass compressing the tissues surrounding the mediastinum and lung parenchyma. Unfortunately, SMARCA4-UT shows a high probability of recurrence after upfront surgery as well as radiotherapy resistance; moreover, chemotherapy has low efficacy. Moreover, given the recent classification of SMARCA4-UT, no data concerning specific clinical trials are currently available. However, several case reports show immunotherapy efficacy in patients with this disease not only in a metastatic setting but also in a neoadjuvant manner, supporting the development of clinical trials. In addition, preclinical data and initial clinical experiences suggest that inhibiting pathways such as CDK4/6, AURKA, ATR, and EZH2 may be a promising therapeutic approach to SMARCA4-UT.

## 1. Introduction

Approximately 5% of conventional non-small cell lung cancers (NSCLCs) are deficient in *SMARCA4* [1]. However, recently, the fifth edition of the World Health Organization (WHO) classification recognized the thoracic *SMARCA4*-deficient undifferentiated tumor (SMARCA4-UT) as a separate entity from conventional NSCLC with *SMARCA4* deficiency because of its distinct histology, immunohistochemical profile, clinical features, and prognosis, placing it under the category of “other epithelial tumors of the lung” [2]. SMARCA4-UT often affects young to middle-aged adults, predominantly males with a history of smoking. Computed tomography (CT) scan-based studies have shown that SMARCA4-UT is a large, invasive mass involving the upper and middle mediastinum, hilum, lung, and/or pleura. Typically, SMARCA4-UTs have strong 18F-fluorodeoxyglucose (FDG) affinity, making positron emission tomography (PET) a useful tool in their clinical staging [3].

Concerning treatment, surgery is effective only for stage I patients because of a high probability of early recurrence [4]. At the same time, SMARCA4-UT patients appear to be resistant to conventional chemotherapy and radiotherapy [5]. Conversely, many case reports and case series have shown the high efficacy of immunotherapy in these patients [1,6,7,8,9]. In addition to immunotherapy, anti-tumor activity in pre-clinical models of *SMARCA4*-deficient tumors has been identified in cyclin-dependent kinase (CDK)4/6, aurora kinase A (AURKA), ataxia telangiectasia-related-3 (ATR), and enhancer of zeste homolog 2 (EZH2) inhibition [10,11,12,13,14,15,16,17,18,19,20,21,22,23,24]. However, because of the recent definition of this clinical pathological entity, no data from randomized controlled trials are currently available, and no guidelines concerning specific SMARCA4-UT treatments have been formulated. Herein, we summarize the biological and clinical features of SMARCA-UT and explore its therapeutic aspects to answer several questions that are still open regarding this new clinical entity: How can we improve SMARCA-UT diagnoses in patients? What is the standard of care for these patients? What is the real incidence of SMARCA4-UT? What type of clinical trials would be best to use for SMARCA4-UT?

## 2. Role of SMARCA4 in Cancer Development

SMARCA4 (BRG1) has a central role in the function of the switch/sucrose-non-fermentable (SW1/SNF) chromatin remodeling complex [25]. This complex modifies gene expression by acting on DNA nucleosome topology, changing chromatin from a dense state to an open state during the gene transcription phase [26,27]. In particular, BRG1 is a catalytic ATPase subunit of the SW1/SNF complex, providing energy for the chromatin remodeling process. As a chromatin remodeler, BRG1 has several biological functions, including embryo development, the differentiation of several types of smooth muscle cells, and spermatogenesis meiosis [28,29,30] (Figure 1a). Moreover, BRG1 has a pivotal role in differentiating myofibroblasts derived from hepatic stellate cells (HSCs) and regulates liver fibrosis. BRG1 is elevated during myofibroblast maturation, while HSC-specific deletion of BRG1 inhibits liver fibrosis [31].

Concerning the role of *SMARCA4* in cancer, BRG1 deficiency is implicated in the development of several malignancies, such as small-cell carcinoma of the ovary hypercalcemic type (SCCOHT), SMARCA4-deficient thoracic sarcoma, colorectal cancer, and endometrial stromal sarcomas [32,33,34,35,36]. Reisman et al. first described the involvement of BRG1 deficiency in lung cancer development [37]. About 10% of lung cancer patients harbor *SMARCA4* mutations, with different clinical features depending on the kind of mutation. In particular, patients with *SMARCA4* homozygous deletions or truncating mutations are prone to early relapse and poor prognoses [38]. BRG1 regulates the expression of several oncogenes and tumor suppressors. Breast cancer gene (BRCA) 1 is a component of the SW1/SNF complex, and BRG1 directly interacts with it, binding BRCA1 with the SWI/SNF complex, thus playing a major role in homology-directed DNA repair [39]. The dominant deletion of BRG1, abrogates p53 expression and thereby mediates BRCA1 [40]. Regarding oncogenes, the proto-oncogenic *MYC* is one of the most relevant. In particular, *SMARCA4* regulates *MYC* activity both via BRG1 binding to *MYC* and via *MYC* target promoters [41]. A lack of BRG1 abrogates the response to retinoic acid and glucocorticoids and upregulates *MYC* targets such as glycolysis-related genes, promoting tumor growth and dedifferentiation [42]. The interaction between BRG1 and c-MYC also regulates adult B cell acute lymphoblastic leukemia (B-ALL) proliferation and apoptosis. In particular, the ectopic expression of BRG1 promotes growth and inhibits the apoptosis of B-ALL, while the downregulation of BRG1 has the opposite effect [43].

*SMARCA4* deficiency also contributes to the cell differentiation block by decreasing Histone3Lysine27 (H3K27) acetylation levels [44]. The modulation of H3K27 activities via BRG1 also plays a major role in human Shh-type medulloblastoma gene regulation [45]. Additionally, in a lung cancer mouse model, *SMARCA4* mutations have induced a high grade of cell dedifferentiation [46]. Concerning tumor suppressor activity, BRG1-containing complexes regulate cellular proliferation by activating retinoblastoma protein through its hypophosphorylation, as mediated by the upregulation of CDK inhibitor p21 [47] (Figure 1b).

The role of *SMARCA4* in tumorigenesis also involves the tumor microenvironment (TME). In particular, M2 macrophage-derived exosomes (MDEs) downregulate BRG1, transferring miR-21-5p and miR-155-5p to cancer cells and promoting their migration and invasion. MDEs contain miR-199a-5p, which also contributes to inhibiting SMARCA4 expression [48,49]. Moreover, BRG1 is a key regulator of cytoskeleton function. *SMARCA4* deregulation results in a decrease in actin stress fibers and thick actin bundles in the cell body, altering cancer cell morphology. At the same time, BRG1 deficiency induces the loss of CD44, a transmembrane glycoprotein implicated in the growth and metastasis of numerous malignancies [50] (Figure 1b).

Nevertheless, the molecular mechanisms of *SMARCA4* in cancer development are not yet fully understood. Concepcion et al. revealed the controversial role of *SMARCA4* in tumorigenesis, with high variability depending on tumor type and *SMARCA4* alterations [46]. For example, decreased *SMARCA4* expression is linked to medulloblastoma and different brain cancers, from low-grade gliomas to glioblastoma. At the same time, *SMARCA4* promotes tumorigenesis in the absence of mutation and through overexpression in other brain tumors [45,51].

A better understanding of the relationship between *SMARCA4* deficiency and cancer development could support the development of new therapeutic strategies.

## 3. Clinical Pathological Features

Recently, the fifth edition of the WHO classification recognized thoracic SMARCA4-UT as a separate entity from conventional NSCLC with *SMARCA4* deficiency because of its distinct phenotype, placing thoracic SMARCA4-UT under the category of “other epithelial tumors of the lung”. Thoracic SMARCA4-UT often affects young to middle-aged adults, predominantly males with a history of smoking accompanied by a genomic smoking signature (Table 1). It is a high-grade malignant neoplasm featuring an undifferentiated or rhabdoid phenotype and BRG1 deficiency [2].

CT-scan-based studies have shown that thoracic SMARCA4-UT is a large invasive mass involving the upper and middle mediastina, the hilum, lungs, and/or the pleura, often resulting in superior vein syndrome, atelectasis, and esophageal invasion, with a primary tumor size of approximately 10 cm [52] (Table 1). Moreover, the associated chest mass appears highly heterogeneous, with marked necrosis and an unclear epicenter [3,52,53,54,55]. In rare cases, the primary lung tumor occurs as a small mass [52,53,54,55]. The most frequent metastatic sites at the time of presentation are the lymph nodes, the adrenal gland, bone, and the lungs, while brain metastases are only rarely described [52,53,54,55]. Symptoms are related to disease extension, including dyspnea, chest pain, fatigue, dysphagia, and weight loss [52,54,55]. Typically, SMARCA4-UT has a strong FDG affinity, making PET a useful tool for clinical staging in addition to CT scans [3] (Table 1). Recently, Shen et al. described a case of SMARCA4-UT that presented as a mediastinal mass and lymph node metastases with intense 68 Ga-DOTA-FAPI-04 uptake, providing new insights into the diagnosis and monitoring of this tumor [56]. Furthermore, if uptake of 68 Ga-DOTA-FAPI-04 can be demonstrated in other SMARCA4-UT patients, new therapeutic approaches might be used for them.

SMARCA4’s histology is characterized by epithelioid and rhabdoid tumor cells with poorly defined nuclear borders and prominent nucleoli. The tumor cells’ growth is organized into non-cohesive clusters without epithelial architecture and with extensive necrosis. Glandular and squamous differentiations are often absent, except in rare cases, combined with juxtaposed NSCLC. The Ki-67 proliferation rate is consistently high at around 70% [57,58,59].

SMARCA4-UT only expresses epithelial markers in some cases in a focal/weak manner, such as cytokeratin (CK) 7, CAM5.2, AE1/3, and epithelial membrane antigen (EMA), while p40, p63, and thyroid transcription factor1 (TTF1) are usually negative. On the other hand, the overexpression of stem cell markers such as sex-determining region Y-Box 2 (SOX2), sal-like protein 4 (SALL4), and CD34 is frequent, as is a loss of caudin-4 expression [60]. SMARCA4-UT is usually negative for desmin, nuclear protein in testis (NUT), S100, and WT-1 [58], but the positive expression of vimentin and synaptophysin has been reported [38]. Focal and weak CD138, CD99, and CD30 expression have also been described [61]. Accordingly, with the high percentage of smoker patients with SMARCA4-UT, there is a high frequency of p53 mutations and an overlap with smoker NSCLC abnormalities, such as mutations of serine/threonine kinase 11 (*STK11*), kelch-like ECH-associated protein (*KEAP1*), and KRAS [55] (Table 2). Moreover, SMARCA4-UT expresses a high tumor mutation burden (TMB) and genome instability because of the DNA repair activity of BRG1. This partly explains the effectiveness of immunotherapy in these patients, as well as cases of predictive mutations related to immune checkpoint inhibitor (ICI) refractoriness [5,54].

**Table 1 ijms-25-03237-t001:** SMARCA4-UT clinical–pathological characteristics.

SMARCA4-UT Features	Description	References
Patient population	Young to middle-aged adults; predominantly male with smoking history.	[2]
Imaging	CT scan shows a large mass involving the mediastinum and lung parenchyma; strong 18F-fluorodeoxyglucose affinity.	[3]
Histopathology	Rhabdoid and/or epithelioid tumor cells organized in non-cohesive clusters without epithelial architecture and with extensive necrosis. High Ki-67 rate of around 70%.	[57,58,59,60,61]
Treatment: Surgery Chemotherapy Radiotherapy Immunotherapy	High rate of recurrence; useful only for stage I. Weak response. Resistance. Promising efficacy	[4,5,6,7,8,9,56,60,62,63]

**Table 2 ijms-25-03237-t002:** Immunohistochemical and biomolecular features.

Methods	Markers	Expression	References
IHC	CK7 CAM5.2 AE1/3 EMA	Only in some cases in a focal/weak manner	[58]
	P40 P63 TTF1 Desmin NUT S100 WT-1	Usually negative	[58,60]
	Vimentin Synaptophysin	Positive in some cases	[38]
	SOX2 SALL4 CD34	Positive	[60]
	Claudin-4	Negative	[60]
	CD138 CD99 CD30	Focal and weak expression	[61]
	SMARCA2	Negative (different from SMARCA4-deficient NSCLC)	[58,61]
	SMARCAB-1	Overexpression	
NGS	P53	81–56%	[64,65]
	KEAP1	41%	
	STK11	39–15%	
	Kras	36–15%	
	NF1	15%	
	PTEN	11%	
	EGFR	Negative	[66]
NGS/IHC/FISH	ROS1	Negative	
	ALK	Some cases reported in the literature	[4,66]

## 4. Differential Diagnosis from Other Malignancies

A correct differential diagnosis between SMARCA4-UT and SMARCA4-deficient NSCLC is necessary for definite prognoses and treatment choices. Regarding rhabdoid and poorly differentiated phenotypes, a lack of epithelial architecture and strong diffuse keratin expression usually excludes SMARCA4-deficient NSCLC. However, differentiation can be difficult in small biopsies, especially if the rhabdoid morphology is lacking [54].

The overexpression of stem cell markers and a lack of adhesion molecule claudin-4 are highly helpful for differentially diagnosing *SMARCA4*-deficient NSCLC. Concerning the other components of the SW1/SNF complex, SMARCA4-UT is often characterized by a loss of SMARCA2 and SMARCAB-1 overexpression [58,61] (Table 2), while *SMARCA4*-deficient NSCLC normally expresses SMARCA2 [52]. There is also a macroscopic difference due to tumor size, which is significantly larger for SMARCA4-UT [58].

However, SMARCA4-UT can mimic other malignancies; with younger-aged patient populations and mediastinum localization, initial diagnosis suspicions are focused on lymphoma and thymic or germ cell tumors. Maartens et al. [62] reported a case of a young man with a large anterior mediastinum mass whose transthoracic biopsy showed uniform malignant epithelioid cells with a clear cytoplasm and a lack of rhabdoid features, mimicking lymphoma. Differential diagnosis was performed via immunohistochemistry (IHC), showing a loss of BRG1 and claudin-4, SALL4 expression, CD34, SMARCAB1, and negative lymphoma markers. Consistent with SMARCA4-UT, the patient presented at an advanced stage at diagnosis with no response to mediastinal radiotherapy [62]. Another study described a case of a young man admitted to the hospital for swelling in multiple lymph nodes and fever. Lymphoma was suspected because of elevated lactate dehydrogenase and soluble interleukin 2 receptor levels as well as bone marrow infiltration from large abnormal cells. This last characteristic can be a sign of advanced disease, as demonstrated by the patient’s death approximately 10 days after admission. The differential diagnosis was obtained via IHC [67].

Concerning sarcomas, a recent case report by Helmink et al. described a SMARCA4-UT with a large iliac bone mass mimicking a primary bone sarcoma with mediastinal lymphadenopathy [68].

Regarding other possible differential diagnoses, a lack of mouse double minute 2 homolog and CDK4 expression excludes liposarcoma. Angiosarcoma shows higher CD34 expression than SMARCA4-UT, expresses diffuse CD31 positivity, and typically exhibits endothelial multilayering. Epithelioid sarcoma is characterized by strong and diffuse vimentin positivity, epithelioid and spindle tumor cells, and cells arranged in cell nodules with a pseudogranulomatous appearance; therefore, it can also be excluded [69,70].

Other *SMARCA4*-deficient malignant tumors, such as malignant rhabdoid tumor (MRT) and SCCOHT, differ from SMARCA4-UT in tumor extension and patient age, occurring mostly in young children. Moreover, SCCOHT typically expresses Wilm’s tumor suppressor gene 1 [71], while SMARCA4-deficient uterine sarcoma, unlike SMARCA4-UT, is not associated with cigarette smoke.

To summarize, a large invasive mass involving the mediastinum, hilum, lungs, and/or pleura in young to middle-aged adults with a history of smoking, characterized by epithelioid/rhabdoid tumor cells, lacking both *SMARCA4* and *SMARCA2* expression, positive for SOX2, SALL4, and CD34, and negative for claudn-4 is a typical presentation of SMARCA4-UT. However, this tumor is not always diagnosed in a straightforward manner, as SMARCA4-UT may mimic other malignancies and IHC features may be misinterpreted [72].

## 5. Treatment: Chemotherapy versus Immunotherapy

Upfront surgery is not a valid therapeutic approach for SMARCA4-UT because of the high probability of early recurrence, except in stage I patients [4]. An observational study comparing resected SMARCA4-UT and conventional NSCLC with SMARCA4 deficiency reported a significantly worse time for progression and OS in SMARCA-UT patients [4]. Furthermore, radiotherapy resistance has been reported in several clinical cases [5].

Despite preclinical data showing *SMARCA4*-knockdown lung cancer cells with high sensitivity to cisplatin because of incomplete repair in both intra-strand adducts and interstrand crosslinks, SMARCA4-UT seems to only have a weak response to conventional chemotherapy [73]. Interestingly, Xue et al. demonstrated that SMARCA4 loss inhibits chemotherapy-induced apoptosis, restricting chromatin accessibility to genes encoding for the Ca^2+^ channel IP3R3 and, as a consequence, impairing Ca^2+^ transfer from the endoplasmic reticulum to the mitochondria required for apoptosis induction [74]. A retrospective study based on a small sample of SMARCA4-UT and NSCLC patients deficient in SMARCA4 showed significantly lower progression-free survival (PFS) using exclusive chemotherapy compared with chemo-immunotherapy (26.8 vs. 2.73 months, *p* = 0.0437) [1]. Similarly, many case reports concerning SMARCA4-UT have demonstrated responses to chemo–immunotherapy or immunotherapy using ICIs as secondary or subsequent treatments after the complete failure of chemotherapy [6,7,8,9]. While one study concerning SMARCA4-deficient NSCLC showed higher chemosensitivity to platinum-based chemotherapy in an adjuvant setting, no study has reported the efficacy of chemotherapy alone in SMARCA4-UT [75]. In the last few years, many studies have demonstrated the efficacy of immunotherapy in SMARCA4-UT patients both as a monotherapy and in combination with chemotherapy [6,7,8,9,63,76].

The biological mechanisms driving the antitumor activity of ICIs in SMARCA4-UT are not yet understood; abnormalities in the SWI/SNF complex correlate with enhanced interferon-gamma-induced T-cell cytotoxicity and better outcomes concerning ICIs in other malignancies [77,78] An enriched Th1 and cytotoxic T-cell microenvironment was recently reported in four cases of SMARCA4-deficient SCCOHT with prolonged responses to ICIs [77]. To the best of our knowledge, only one study has reported a low response rate to ICIs in SMARCA4-UT, which was associated with an absence of tumor-infiltrating lymphocytes in the TME [79] Conversely, Hozumi et al. [80] showed that low SMARCA4 expression correlates with the upregulation of the T cell effector; the mature B cell marker; and central memory marker genes, such as CD8B, CD40LG, CD20, CD38, CD79, interferon Regulatory Factor 4, CD27, and -C motif chemokine receptor 7. At the same time, chemokines that attract functional T and B cells inside tumors (namely, chemokine (C-C motif) ligand (CCL)19 and CCL21) are upregulated, while chemokines that induce an immunosuppressive TME (namely, Interleukin 20 receptor, alpha subunit, CD200, and CXCL8) are downregulated in patients with cancer harboring low SMARCA4 expression. Overall, low *SMARCA4* expression correlates with an upregulated tertiary lymphoid structure (TLS)-associated gene signature, suggesting that the development of mature B cell accumulation in SMARCA4-deficient tumors is a potential mechanism of immune checkpoint-based therapy sensitivity [80].

Moreover, immunotherapy efficacy in SMARCA4-UT may be explained by genome instability due to a lack of BRG1. BRG1 promotes DNA repair at double-stranded breaks and the recruitment of repair factors [81]. Immunotherapy efficacy in SMARCA4-UT seems to be independent of programmed death-ligand 1 (PD-L1) expression, which is often low or negative [82,83]. For example, Naito et al. described a complete response in SMARCA4-UT patients treated with nivolumab after three rounds of standard chemotherapy, with no PD-L1 expression [8]. Similarly, Yang et al. reported that they successfully treated a PD-L1-negative SMARCA4-UT patient with a second-line regimen containing tislelizumab, etoposide, and carboplatin after the failure of first-line chemotherapy consisting of four cycles of liposomal paclitaxel and cisplatin [5]. However, both patients exhibited a high TMB. Moreover, in SMARCA4-UT patients, immunotherapy responses can be extremely rapid. Shi et al. reported a case of SMARCA4-UT with an Eastern Cooperative Oncology Group performance status (ECOG) of 3 that was unresponsive to chemotherapy because of dyspnea, thoracic pain, and hemoptysis. The patient obtained clinical benefit with the resolution of respiratory symptoms after a one-dose tislelizumab (200 mg) infusion, reducing their ECOG score from 3 to 1 [7]. Nevertheless, the high efficacy of immunotherapy in SMARCA-UT has been reported in many cases, so prospective studies are urgently needed to define its correct use in these patients.

Interestingly, some cases have also shown the efficacy of immunotherapy as a conversion surgery treatment. Two SMARCA4-UT patients underwent mediastinal mass resection after treatment with ICIs plus chemotherapy, achieving a complete pathologic response [78]. Kunimasa et al. reported a case of SMARCA4-UT with chest wall and vertebral invasion that was successfully treated with conversion surgery after a neoadjuvant therapy consisting of atezolizumab in combination with bevacizumab, paclitaxel, and carboplatin [76]. Another case report described a surgical resection after the failure of second-line chemotherapy using a combination of ipilimumab and nivolumab. The resection was radical, with a complete pathological response [79].

Recently, Wang et al. described the long-term benefits of pembrolizumab in patients with undifferentiated carcinomas of the gastrointestinal tract with SMARCA4 deficiency, contributing to a growing body of evidence for the efficacy of ICIs in patients experiencing SMARCA4 alterations [84].

## 6. Treatment: Novel Targets

A loss of BRG1 results in the profound downregulation of cyclin D1 (CCND1), both because of restricted CCND1 chromatin accessibility and the suppression of a transcription activator of CCND1, c-Jun. Xue et al. demonstrated that SCCOHT cells are characterized by limited CDK4/6 kinase activity and increased susceptibility to CDK4/6 inhibitors [85], suggesting that CDK4/6 inhibitors approved for breast cancer subtypes could be repurposed to treat SMARCA4-deficient tumors. Interestingly, a patient with SMARCA4-deficient SCCOHT was reported to have achieved a good response to a combination of abemaciclib and nivolumab despite experiencing tumor growth through multiple prior regimens, including chemotherapy, radiation therapy, and immunotherapy with both anti-PD-1 and anti-CTLA-4 therapy [10]. Subsequently, the same authors confirmed this susceptibility to CDK4/6 inhibition in NSCLC with SMARCA4 expression loss. Moreover, SMARCA2 loss also contributes to inhibited CCND1 expression, making CDK4/6 inhibitors a promising treatment for SMARCA4-UT, as they are often characterized by a loss of both SMARCA4 and SMARCA2 expression [86]. However, in a recent nonrandomized, multidrug, pan-cancer trial with tumors harboring cyclin CDK4/6 pathway alterations, including SMARCA4 expression loss, only limited clinical activity for palbociclib and ribociclib monotherapy was observed [87]. AURKA expression is often elevated in several cancer types, and a link between SW1/SNF complex dysfunction and AURKA overexpression has been reported for rhabdoid tumors. Preclinical data have shown that the loss of BRG1 increases sensitivity to AURKA inhibitors [11]. Further studies targeting AURKA activity in SMARCA4-UT are needed.

EZH2 is the catalytic subunit of polycomb repressive complex 2 (PRC2), which is responsible for H3K27 methylation [12]. Anomalous levels of PRC2 have been observed in SMARCB1 mutation and SMARCA4 mutation cancers [13,14], with a natural antagonistic relationship between PRC2 and the SWI/SNF complex [15]. Tazemetostat, a selective EZH2 inhibitor, has inhibited cell growth and induced apoptosis and differentiation in many preclinical studies concerning MRT, lymphoma, and SMARCA2/SMARCA4-negative small-cell carcinoma of the ovary with hypercalcemia [16,17]. In a phase I/II trial (NCT01897571) including lymphoma and advanced solid tumors, tazemetostat showed clinical benefit in patients with SMARCA4 loss [18]. More recently, the NCI-COG Pediatric MATCH APEC1621C trial—which includes patients with actionable alterations such as SMARCB1 loss, EZH2 mutation, and SMARCA4 loss—showed an objective response in a patient with SMARCA4-deficient non-Langerhans cell histiocytosis [19]. Regarding immunotherapy efficacy in SMARCA4-deficient cancers, a trial evaluating the combination of nivolumab/ipilimumab and tazemetostat is ongoing [20]. Interestingly, PRC2 is a therapeutic target of the TopoII inhibitor etoposide, and as a consequence, SMARCA4 loss may predict sensitivity to etoposide in response to EZH2 inhibition [21].

ATR is involved in many DNA repair pathways, acting on single- and double-stranded DNA damage [23,24]. Intriguingly, a loss of BRG1 in lung cancer cells leads to the activation of replication stress responses and ATR dependency, suggesting that ATR inhibition could be a potential therapeutic approach to SMARCA4-deficient tumors [24].

Finally, SMARCA4 mutant tumors express the signature of enhanced oxidative phosphorylation (OXPHOS). Accordingly, inhibiting OXPHOS with small-molecule IACS-010759 results in high efficacy in *SMARCA4* mutant lung cancer cell lines and xenograft tumors [88]. Regrettably, two phase I trials demonstrated limited antitumor activity at tolerated doses, reporting elevated blood lactate levels and neurotoxicity [64,89]. Additional studies are needed to evaluate the efficacy and safety of these molecules.

Moreover, considering the relationship between BRCA1 and BRG1, the use of PARP inhibitors should be evaluated in clinical trials [90].

Unfortunately, no data concerning clinical trials on SMARCA4-UT are currently available. Studies that can identify biomarkers for predicting responses to immunotherapy and clinical trials exploring the combination of ICIs with the abovementioned pathway targeting are needed.

## 7. SMARCA4-UT and Co-Occurring Mutations

Considering the higher percentage of smokers and ex-smokers in the SMARCA4-UT population, it should not be surprising that the most frequent co-occurring mutations in these patients are similar to those detected in NSCLC patients who are also smokers. Schoenfeld et al. evaluated the genomic profiles of 4813 tumors from patients with lung cancer, identifying 407 patients harboring SMARCA4 alterations. In these patients, the most frequent co-mutations were TP53 (56%), KEAP1 (41%), STK11 (39%), and KRAS (36%), with STK11 and KEAP1 showing a stronger association with SMARCA4 alterations [64]. In particular, STK11 mutation occurred more frequently in patients with SMARCA4 truncating mutations, gene fusions, and homozygous deletions, characterized by a higher frequency of BRG1 loss. Controversially, these SMARCA4 alterations correlated with a higher response rate to ICIs, suggesting a role for predictive factors independent of the STK11 state [91,92].

Conversely, KRAS mutations in patients with SMARCA4 alterations lead to a lower efficacy in immunotherapy as a monotherapy and as a combination with chemotherapy. Translational studies have shown that KRAS and SMARCA4 co-mutation correlates with a lower proportion of CD8+ T cells and CD4+ memory T cells [93].

At the same time, the co-existence of SMARCA4 alterations and KRASG12C impairs clinical outcomes using KRAG12C inhibitor monotherapy [94]. However, these data come from studies mainly focused on SMARCA4-deficient NSCLC, which has a distinct overlap with SMARCA4-UT but is not the same clinicopathologic entity.

Recently, Wang et al. characterized SMARCA4-UT patients using 68-panel next-generation sequencing (NGS), confirming TP53 as the most frequent co-mutation (81%), followed by CDKN2A (26%), KRAS (15%), STK11 (15%), NF1 (15%), and PTEN (11%) [65].

Targetable co-mutations such as epidermal growth factor EGFR, ALK, and ROS proto-oncogene 1 are infrequent in SMARCA4-UT patients. Sheng et al. reported a case of SMARCA4-UT with EML4-ALK fusion, which was successfully treated with alectinib [66]. More recently, an observational study including SMARCA-UT patients and SMARCA4-deficient NSCLC reported three cases of patients with ALK mutation, including one EML4-ALK rearrangement [4]. Further studies are needed to define the role of targetable co-mutations in SMARCA4-UT. Moreover, studies exploring the role of co-mutations focusing exclusively on SMARCA4-UT patients are needed.

## 8. Future Perspectives

SMARCA4-UT is a separate entity from conventional NSCLC with SMARCA4 deficiency and has different clinical pathological features and sensitivities to treatments. Given the peculiarity of this thoracic cancer and its recent definition in the WHO classification, a multidisciplinary approach to its diagnosis that includes clinical, radiologic, and pathologic factors is essential [2]. Nevertheless, even today, some cases of SMARCA4-UT may be misdiagnosed since a specific immunohistochemistry analysis for this tumor is not routinely performed. On the other hand, no specific guidelines for SMARCA4-UT diagnosis and treatment have been formulated. Therefore, the major questions regarding SMARCA4-UT patients are as follows: How can we improve SMARCA-UT diagnosis? Can we consider immunotherapy a standard of care for these patients?

Cases of SMARCA-UT may be misinterpreted as pulmonary pleomorphic carcinoma or other cancers involving the mediastinum, lungs, and pleura. Therefore, in cases of clinical features suggesting SMARCA4-UT, implementing an IHC panel with SMARCA4, SMARCA2, SOX2, SALL4, CD34, and claudn-4 is mandatory [72].

Concerning treatment, our findings show that immunotherapy may be helpful, and it should be used as a first-line treatment. Moreover, cases of SMARCA4-UT conversion surgery after immunotherapy have also been described [76,78,79]. Conversion surgery is an interesting therapeutic option for SMARCA4-UT patients, who tend to be young or middle-aged. It is also a tumor extension that is often characterized by an invasive mass involving the upper and middle mediastina with only tardive metastases. At the same time, several case reports have demonstrated the efficacy of ICIs in SMARCA4-UT patients independent of PD-L1 expression [8,82,83], suggesting that this biomarker may not be suitable for clinical decision making in these patients.

Further studies exploring the role of ICI monotherapy or ICIs plus chemotherapy in SMARCA4-UT are needed, as are studies focused on predictive factor research.

Other crucial questions concerning SMARCA4-UT are as follows: What is the real incidence rate of SMARCA4-UT? Are the number of SMARCA4-UT patients suitable for classical randomized trials? Is there a role for basket trials?

Given the recent description of this clinical pathological entity and considering the potential rate of misdiagnosis, the real incidence of SMARCA4-UT may be higher than reported in the literature, thus enabling the development of ad hoc clinical trials. On the other hand, the use of basket trials could also be considered, resulting in the faster recruitment of patients with malignancies harboring SMARCA4 deficiency. For example, similar to SMARCA4-UT, SCCOHT, gastrointestinal tract SMARCA4-deficient undifferentiated tumors, and other cancers harboring SMARCA4 downregulation have shown sensitivity to immunotherapy [10,84]. Moreover, further studies exploring the inhibition of pathways such as CDK4/6, AURKA, ATR, and EZH2 are anticipated.

## 9. Conclusions

Considering the young age of SMARCA4-UT patients and this disease’s low response rate to chemotherapy, clinical trials specifically designed for SMARCA4-UT are urgently needed. At the same time, given the peculiarity of SMARCA4-UT, it is important to increase the sensitivity of oncologists to other lung tumors during differential diagnosis, enabling the best possible therapeutic decision making.

## Figures and Tables

**Figure 1 ijms-25-03237-f001:**
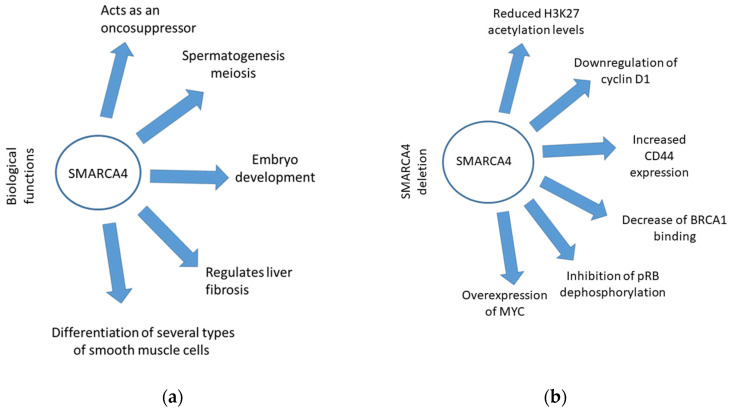
Effects of SMARCA4 and its deletion: (**a**) summary of the biological functions of SMARCA4; (**b**) summary of all pathological conditions related to SMARCA4 deletion.

## Data Availability

Not applicable.

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
