# Peer review of "Treatment of Thoracic SMARCA4-Deficient Undifferentiated Tumors: Where We Are and Where We Will Go"

_ijms, 2024, doi:10.3390/ijms25063237_

Round 1

Reviewer 1 Report

Comments and Suggestions for Authors

>> How has the recognition of thoracic SMARCA4-deficient undifferentiated tumor (SMARCA4-UT) as a distinct entity from conventional NSCLC evolved over the last few years, and what are the key clinicopathological characteristics that differentiate them?

>> What demographic trends are observed in patients with SMARCA4-UT, particularly regarding age groups and the anatomical locations within the thoracic region?

>> Could you elaborate on the challenges associated with upfront surgery, radiotherapy, and chemotherapy in the treatment of SMARCA4-UT, including the high probability of recurrence and resistance to standard therapies?

>> What evidence and insights do case reports provide regarding the efficacy of immunotherapy in patients with SMARCA4-UT, both in the metastatic setting and as a neoadjuvant approach?

>> Considering the recent classification of SMARCA4-UT, how does the absence of specific clinical trial data impact the current understanding and treatment options for this condition?

>> What specific factors or characteristics of SMARCA4-UT make immunotherapy an effective option, and what is the rationale behind considering immunotherapy in a neoadjuvant manner?

>> Could you provide insights into the emerging therapeutic approaches for SMARCA4-UT, including the inhibition of pathways such as CDK4/6, AURKA, ATR, and EZH2? What preclinical data or initial clinical experiences support these approaches?

>> How do the identified pathways—CDK4/6, AURKA, ATR, and EZH2—hold promise as therapeutic targets for SMARCA4-UT, and what mechanisms underlie their potential efficacy?

>> While specific clinical trial data for SMARCA4-UT may be lacking, are there indications or ongoing trials that suggest a potential role for immunotherapy in the broader landscape of thoracic tumors?

>> Based on the current status of knowledge, where do you envision the future of SMARCA4-UT treatment heading, and what research gaps need to be addressed to enhance our understanding and improve therapeutic outcomes?

Comments on the Quality of English Language

The language usage throughout this paper need to be improved, the author should do some proofreading on it. 

Author Response

Dear Reviewer, I attached a word file, with responses  to your comments and suggestions. Thank you for helping improve our manuscript. 

Reviewer 2 Report

Comments and Suggestions for Authors

After reviewing the manuscript titled "Treatment of Thoracic SMARCA4-deficient undifferentiated tumor: where we are and where we'll go," I have several concerns that need to be discussed with the authors.

1. The authors are advised to carefully review the manuscript to avoid any unnecessary errors. For instance, in the "Author Contributions" section, the authors have not removed the template text.

2. The use of numerous acronyms is observed; however, there is no index for these acronyms.

3. It is suggested to separate the biological functions of SMARCA4 and its relationship with tumors in Figure 1 to facilitate easier reading for the audience.

4. The authors are recommended to supplement necessary figures and tables to assist readers in understanding and grasping the core of the article.

Author Response

Dear reviewer, 

   Thank you for helping improve our manuscript. We have carefully reviewed the paper. In particular, we removed the templete text in the Author Contributions section. Moreover, we added a abbreviations paragraph. 

Concerning to figure 1, we divided the picture in two parts, concerning biological functions and the link with cancer of SMARCA4. We also added a new table,, namelly table 2 concerning histochemical and biological features of SMARCA4-UT.

Reviewer 3 Report

Comments and Suggestions for Authors

The article prepared by Vito Longo et al. is a review-type article regarding the biological, clinical and therapeutic features of SMARCA-UT.

To improve the quality of the manuscript, I recommend the following:

- Kindly revise Figure 1 to improve its topographical quality for readers.

Please explain all the abbreviations used in Figure 1 in its legend.

Please revise the lines 114-115 and 118-119 – they are almost similar.

- Kindly add a reference according to the data presented in line 122.

- Please add histopathological photo(s) of SMARCA4-UT in order to impact us.

- Please summarize the immunohistochemical data (lines 140-152) described in a Table for readers.

- Kindly add the limits and a Conclusion section of this study.

Please revise the references list according to the Int. J. Mol. Sci. journal recommendations.

Author Response

Dear reviewer, 

   Thank you for help to improve our manuscript. We have revised the Figure 1 dividing it in two picture, one concerning the biological functions and the other about pathological conditions supported by SMARCA4 deficency. 

We revised line 114-115 and 118-119, excluding the similarities. 

We have added a reference concerning the line 122.

We summarized immunoistochemical data in a new table, namely table 2.

We have  added a conclusions section

Finally, we revised the refernces list according to JIMS, 

Best regards

Round 2

Reviewer 2 Report

Comments and Suggestions for Authors

I don't have any more comments after the author's revisions.

Author Response

Thank you very much for your help to improve our manuscript.

Reviewer 3 Report

Comments and Suggestions for Authors

The authors provided a revised version of the manuscript addressing most comments. Before recommending acceptance, I recommend the references list revision according to the Int. J. Mol. Sci. journal instructions. Kindly see the following link: https://www.mdpi.com/journal/ijms/instructions

Author Response

(The authors gave the same response as above.)
